# Determination of the Predictors with the Greatest Influence on Walking in the Elderly

**DOI:** 10.3390/medicina58111640

**Published:** 2022-11-13

**Authors:** Chul-Min Chung, Sunghoon Shin, Yungon Lee, Do-Youn Lee

**Affiliations:** 1Research Institute of Human Ecology, Yeungnam University, Gyeongsan 38541, Republic of Korea; 2Neuromuscular Control Laboratory, Yeungnam University, Gyeongsan 38541, Republic of Korea

**Keywords:** gait, muscle strength, static balance, body composition

## Abstract

*Background and Objectives:* Previous studies have revealed that independent variables (lower extremity strength, postural control ability, and body composition) influence gait performance and variability, but the difference in the relative influence between these variables is unclear. Hence, this study determines the variable that is the most influential predictor of gait performance and variability among potential independent variables in the elderly. *Materials and Methods:* Seventy-eight subjects aged ≥60 years participated. For each subject, the gait variables and lower extremity muscle strength were measured using an accelerometer worn on both feet during a 6-minute walk and a manual force sensor, respectively. The static balance ability was measured through two force plates, and the body composition was measured by applying bioelectrical impedance analysis. Linear regression analyses were performed stepwise to determine whether these variables affect gait performance and variability. *Results:* After adjusting for sex and gait performance, the ankle strength, body fat mass, mean velocity in the medial–lateral direction, ankle plantar flexion strength, and girth were predictors of gait speed dorsiflexion, gait performance, swing width of the gait performance, walking speed, and gait variability, respectively. *Conclusions:* Overall, gait performance in the elderly is related to muscle strength, postural control, and body composition in a complex manner, but gait variability appears to be more closely related to ankle muscle strength. This study provides further evidence that muscle strength is important in motor function and stability.

## 1. Introduction

Walking is an essential movement that is constantly repeated in daily life [1]. In general, gait ability is quantified as the average of the spatiotemporal variables during gait (i.e., walking velocity, stance duration, swing duration, double support, stride length, step width, etc.), and gait variability is the standard deviation or fluctuation of the gait variables expressed as a coefficient [2,3,4,5,6,7]. The evaluation of gait performance (gait ability) and gait variability is used to quantify the stages of aging and movement disorders, and to verify and improve the effectiveness of therapeutic interventions, exercise, and rehabilitation programs [8]. In particular, in gait research targeting the elderly, gait ability and variability are used to evaluate the changes in the dynamic stability due to aging, deterioration of physical function, and the presence or absence of movement disorders [5,9,10,11,12].

Looking at the characteristics of the temporal and spatial variables of gait, it is understood that a decrease in the average gait speed increases the likelihood of a fall in the elderly [13,14], indicates decreased physical function, and is associated with a high mortality rate [15]. In general, walking speed tends to decrease with a decreasing stride length as age increases [16,17]. Its decrease is also correlated with age, lack of physical activity, decreased physical function, and cardiovascular disease in the elderly [18]. Based on these findings, previous studies have predicted the risk of falls [16,17]. In particular, short stride length and both excessively high or low step width variability in the elderly are known to increase the risk of falling [4,6,19]. Conversely, an increase in double support and stance duration during gait were found to be common features used as a compensation strategy for a stable gait, which reduces the risk of falls in healthy elderly people [20,21].

Investigation into the potential factors that can be used to predict the changes in the gait variables strengthens the value of functional gait research in the elderly and opens up the possibility of reducing the risk of falls through improved gait [22,23,24,25]. Therefore, the contribution of the potential influencing factors toward walking ability should be evaluated. The major factors known to affect walking in the elderly in previous studies on walking are as follows.

First, good lower extremity muscle strength is emphasized as a potential factor that can improve the walking performance of the elderly and reduce the risk of falls [26]. It has been reported that elderly individuals have reduced gait performance because the muscles of the hip, knee, and ankle of the lower extremities are weakened [27,28,29]. For example, Anderson and Madigan [27] investigated whether hip and plantar flexor torque and the corresponding estimate of range of motion are limiting factors in the gait of the elderly. They argued that the elderly do not exert sufficient plantar flexor torque when walking compared to young adults [27], and show reduced gait performance due to weak hip extension [30]. Muehlbauer et al. [29] also analyzed the relationship between the lower extremity strength and gait performance in the elderly, and reported that hip extension and ankle plantar flexion strength predict walking speed and stride length. Shin et al. [7] investigated the relationship between knee extension strength and variability of the step width during gait, and reported that improvement in muscle strength can help prevent gait disorders and falls.

Second, impaired static postural control has been reported to reduce single support phase time during walking and induce asymmetry of the left and right lower extremities, resulting in irregular gait [31,32]. However, Shimada et al. [33] reported that static posture control ability in the experimental treatment with different tactile and visual environments was independent of both factors, and not related to walking on the treadmill. Clarifying the importance of postural control can clarify whether the contradictory results of the previous studies are due to the differences between actual walking and treadmill walking, or whether the effect of postural control ability at rest on gait is an insignificant factor affecting gait performance.

Third, it has been reported that increased fat and decreased muscle mass in the elderly deteriorate gait and postural control, causing functional impairment [34,35]. Woo et al. [36] showed that fat mass is a major factor influencing walking speed. In addition, LaRoche et al. [37] reported that fat mass limits gait performance and has an explanatory power of 42% for the maximum speed during walking. Ko et al. [38] reported that compared with normal-weight elderly individuals, obese elderly individuals had a higher knee flexion duration, and that gait speed decreased as the BMI increased. Based on these results, it is highly likely that the body composition factor is a major latent variable affecting gait ability and variability among the elderly [35]

Fourth, chronic lower body pain causes gait differences in the elderly [39]. De Kruijf et al. [38] found that elderly individuals with chronic pain in the leg and hip walk with lower rhythm, phases, and pace; slower and smaller steps; longer double support; and more asymmetry, which is independent of osteoarthritis.

Fifth, the elderly have insufficient hip extension range of motion (ROM) for walking compared to younger adults [27]. Anderson and Madigan [27] showed that elderly people walk with less pelvic rotation ROM at both comfortable and fast walking speeds, and less trunk rotation ROM at fast walking speeds. Kang et al. [40] showed that decreased ROM in healthy elderly individuals could independently be a major clinical predictor of increased instability and decreased motor function, even if their muscle strength is not reduced enough to decrease gait speed. 

Previous studies have established that lower extremity muscle strength, postural control ability, and body composition influence gait performance, defined as the average values of spatiotemporal gait parameters and gait variability, quantified as the standard deviation or coefficients of variance of gait parameters; however, the difference in the relative influence between these variables is unclear. That is, assuming that these variables simultaneously affect the dependent variable, i.e., gait (quantified by gait performance and variability), it is not known which variable is the most important influencing factor. Therefore, this study aimed to determine the variable that is the most influential predictor of gait performance and variability among the potential independent variables affecting gait in the elderly. 

## 2. Materials and Methods

This study was approved by the Yeungnam University Bioethics Committee, and all participants provided written informed consent.

### 2.1. Participants

A total of 78 subjects (males, 38; females, 40) over the age of 70 years participated in this study (Table 1). Participants were recruited from a pool of healthy seniors living in the community through community notices. Participants were selected based on their ability to walk independently and their willingness to engage in physical activity. The average age of the subjects was 75 years (75.69 ± 5.95), and only those who had no musculoskeletal diseases or neurological pathological symptoms in the last 6 months were selected; we excluded patients with all types of COPD and dyspnea. In addition, individuals with neuromuscular diseases who had artificial joints or metal devices inserted were excluded from the study. All participants participated in the study after providing written consent for the purpose, procedure, and precautions of the study. The physical characteristics of the subjects and information on the measured muscle strength, static posture control, body composition, and gait performance are presented in Table 2.

### 2.2. Study Protocol

Every subject underwent a 6-minute walk test, a static balance test, and body composition measurement, which were conducted in random order (Figure 1). Lower extremity muscle strength was measured last to avoid muscle fatigue influence.

#### 2.2.1. Lower Extremity Muscle Strength Test

Lower extremity muscle strength was measured using microFET3 (Hoggan Health Industries, West Jordan, UT, USA). MicroFET3 is a portable force sensor that measures the maximum force based on the pressure gauge of the dynamometer, and the measured value can represent a force of up to 890 in Newton (N) units. In a previous study, the reliability of the sensor to measure isometric force was verified [41]. The main movements performed to measure the lower extremity strength were flexion iliopsoas, extension gluteus, abduction gluteus, flexion hamstring, extension femoris/vastus, plantar flexion, and dorsiflexion. The lower extremity strength data were collected on the basis of previous studies [27,28,29]. All movements were performed on a massage bed, and the muscle strengths of both lower extremities were measured according to the manufacturer’s instructions. Subjects were asked to measure the muscle strength of each extremity three times, with maximum force measured for 3 s. The rest time between measurements was set at 10 s, and each measurement was divided into 30 s of rest time to avoid fatigue after three measurements. The maximum values of the three measurements for each movement were used for the data analysis. The measurements were performed by one person to ensure consistency. The specific muscle strength test process was as follows. Subjects in the Flexion iliopsoas test lifted their thighs from a seated position on a message bed. Participants performed gluteus maximus extension in a prone position with the hip joint in neutral and the knee joint flexed 90°. Participants in the gluteus maximus abduction exercise were asked to lie on their sides and raise their legs while keeping their knees straight. The sensor was supported on the lateral malleolus by the researcher. Participants in the hamstring flexion exercise were prone, with the hip joint in neutral and the knee joint flexed 90°. The sensor was placed on the subject’s heel, and the participant was instructed to flex their knee as far beyond a 90-degree angle as possible. In the supine position, the participants extended the knee and dorsiflexed the ankle as much as possible. During dorsiflexion, the researcher supported the subject with sensors. A sensor was placed on the metatarsal head dorsal surface. In the supine position, the researcher measured plantar flexion of the ankle while extending the knee and extending the plane of flexion as much as possible. A sensor was placed on the metatarsal head plantar surface.

#### 2.2.2. The 6-Minute Walk Test

The 6-minute walk test is highly correlated with functional performance in the elderly [42], and is a valid and reliable measurement method [43]. The 6-minute gait was measured during self-paced walking selected by the subjects in the indoor gym. The participants wore two Inertial Measurement Unit sensors (Physilog5^®^, GaitUp™, Lausanne, Switzerland) in the instep of their shoes. Right foot data were used in this study. Walking distance was set to 30 m. The turning point was marked by installing a cone at the starting point and at the end of the 30 m point. The mean number of cycles (±SD) was 353 ± 20.76. The subject repeatedly turned between both ends of the 30 m distance for 6 min. To collect kinematic data measured from the sensors attached to the instep, the manufacturer’s software, installed on a Lenovo T520 laptop (Lenovo Group Ltd., Beijing, China), was used. Gait variables such as speed, double support, mean values of the spatial variables (stride length and swing width), and the coefficients of variation were extracted. The gait variables, as dependent variables, were calculated with “Gaitup lab” software, which was developed by the manufacturer. The average value and coefficient of variation of the gait variables were defined as walking performance and gait variability, respectively. Each set of gait variables was selected based on the variables used in previous studies [4,6,19,20,21]. The turning period was based on the turning angle; if the turning angle was over 20°, the period was excluded from the analysis [44].

#### 2.2.3. Static Balance Test

Two force plates (Accusway, Advanced Mechanical Technology, Inc, Watertown, MA, USA) and a data acquisition system (PXIe-1078/6363, National Instruments, Austin, TX, USA) were used to measure forces (Fx, Fy, and Fz) and moments (Mx, My, and Mz) at the sampling frequency of 100 Hz. Participants stood with one foot each (left and right) on two force plates in a natural static upright posture at shoulder width, and were asked to gaze at a fixed target at eye level, 2 m in front of them. The recorded data were filtered using a fourth-order low-pass Butterworth filter with a cutoff frequency of 10 Hz, and then the center of pressure (COP) in the anterior–posterior (AP) and medial–lateral (ML) directions was calculated from two force plates. The net COP from the two plates was extracted from two force plates using to the following equation [45]:$$\mathrm{COP}\;\mathrm{net}=\mathrm{COPl}\frac{\mathrm{Rvl}}{\mathrm{Rvl}\;+\;\mathrm{Rvr}}\;+\mathrm{COPr}\frac{\mathrm{Rvr}}{\mathrm{Rvl}\;+\;\mathrm{Rvr}}$$
where COPl is the left COP; COPr is the right COP; and Rvl and Rvr are the vertical reaction forces from the left and right feet, respectively. The mean velocity and root mean square (RMS) of COP signals in both the AP and ML directions were considered as static balance variables. This is because mean velocity showed the highest reliability among trials, and RMS is related to the variability of postural control dynamics [46].

#### 2.2.4. Body Composition Measurement

Body composition was measured using an InBody bioelectrical resistance device (InBody 520, Biospace Co., Ltd., Seoul, Korea). Bioelectrical impedance analysis (BIA) is a convenient method for estimating body components such as skeletal muscle mass, body fat mass, minerals, and body water through bioelectrical signals [47]. BIA has been evaluated as an objective and reliable method [48]. In this study, the soft lean mass, body fat mass, and BMI were extracted from the bioelectrical signals estimated using the BIA, and were used as independent variables. Each variable was selected by referring to previous studies that evaluated the differences in walking ability through changes in body fat mass, muscle mass, and BMI [34].

### 2.3. Data Analysis

In the Spearman correlation analysis, after extracting the independent variable that showed a significant correlation with the dependent variables, gait performance and gait variability, in order to avoid multicollinearity, variance inflation factors (VIFs) were estimated for all independent variables. Lower extremity muscle strength during flexion of the iliopsoas, extension gluteus, abduction gluteus, flexion hamstring, extension femoris/vastus, plantar flexion, and dorsiflexion were selected as lower extremity strength variables. RMS_AP, RMS_ML, MV_AP, and MV_ML were selected as static balance variables. Soft lean mass, body fat mass, and BMI were selected as body composition variables. Height and sex [49] were the control variable and moderator, respectively. Therefore, a total of 16 variables were input as independent variables, and gait speed, gait temporal variables (stance, swing, and double support duration), and spatial variables (stride length and swing width) were used as dependent variables. At this time, the normality, independence, and equal variance of each variable were confirmed. All data were converted to z-values and statistical analyses were performed.

Stepwise linear regression analyses were repeated to determine whether the lower extremity muscle strength, static posture control, and body composition variables affected gait performance for each dependent variable of gait performance and variability: speed, stance, double support duration, stride length, swing width, and CVs of each variable. After stepwise regression analysis, the significant results were adjusted with Holm correction to reduce type I error [50]. All statistical analyses were performed using SPSS version 20 (IBM Inc., Chicago, IL, USA). The significance level was set to <0.05. G*Power software was used to perform a priori power analysis in order to determine the sample size during multiple regression analysis for the validation of the measurement model. The required power was set at 1 − β = 0.85. Level of significance was set at α = 0.05. Effect size was set with a large range value of 0.35. The number of predictors was 16. Using these settings, a power curve was extracted. A total sample size of *n* = 76 was needed for this study. 

## 3. Results

### 3.1. Prediction of Walking Speed from the Lower Extremity Strength, Posture Control, Body Composition Factors

In the stepwise linear regression analysis of the lower extremity strength, static balance, and body composition factors against walking speed, one variable out of 16 independent variables predicted gait speed, and gait speed CV (Table 3 and Table 4). In the case of lower extremity strength, ankle dorsiflexion strength influenced the gait speed (r² = 0.291), and ankle plantar flexion strength affected the gait speed CV (r² = 0.426).

### 3.2. Prediction of Temporal Gait Parameters of Gait from the Lower Extremity Strength, Static Balance, and Body Composition Factors

In the stepwise linear regression analysis of the lower extremity strength, static balance, and body composition factors against temporal gait parameters (stance duration, swing duration, and double support duration), none of the 16 independent variables were significantly predicted in the regression model upon conducting Holm correction (Table 3 and Table 4).

### 3.3. Prediction of Spatial Gait Parameters from the Lower Extremity Strength, Static Balance, and Body Composition Factors

In the stepwise linear regression analysis of lower extremity muscle strength, static balance, and body composition factors against spatial variables of gait (stride length and swing width), one of the 16 independent variables influenced the swing width CV (Table 3 and Table 4). Among the lower extremity strengths, ankle plantar flexion strength affected swing width CV (r² = 0.190).

## 4. Discussion

This study investigated the relationship between the lower extremity muscle strength, static balance, body composition, gait performance, and gait variability in elderly individuals aged 60 years or older. To determine the magnitude of the influence of the independent variables that could potentially affect the walking ability, a step-by-step regression analysis was performed by classifying the dependent variables, gait performance and variability, walking speed, and spatial and temporal variables. Confirming the influence of independent variables on the walking performance of the elderly is expected to be valuable as important clinical information for effectively improving the walking ability of the elderly, with large differences in their individual mechanical and physiological physical abilities, and in preventing falls [51,52].

Stepwise linear regression analysis indicated the variables that preferentially affect the gait performance from among the independent variables. The main findings of this study are as follows. First, ankle dorsiflexion strength had the most significant effect on walking speed in the elderly. A reduction in lower extremity muscle strength has been suggested as a major factor inducing falls by reducing mobility and stability in daily life [53,54]. Previous studies have shown that knee and ankle strength in the elderly are related to walking speed, highlighting the importance of lower extremity strength in improving walking speed, the lack of which is the cause of falls [55]. Burnfield et al. [30] confirmed the relationship between the maximum isokinetic torque of the hip joint, knee, and ankle in the elderly and the preferred gait speed during a 10 m walk using regression analysis, showing that the hip extension torque is the only predictor of gait speed. In this study, however, ankle dorsiflexion strength had the only effect on walking speed in the elderly during the 6-minute walk test. This difference from previous studies is thought to be because the influence of muscle strength on walking speed for each lower extremity segment acted at different ratios depending on the walking distance and time during the actual experiment. In addition, contrary to the findings of this study, previous research has found a link between ankle plantar flexion strength and preferred and maximum gait speed in the elderly. Muehlbauer et al. [29] showed the relationship between the hip, knee, and ankle muscle strength, and maximum and preferred gait speed in the elderly. Plantar flexion strength, evaluated as the ankle muscle strength, showed a significant relationship with the maximum gait speed, and was correlated with the preferred gait speed. It was shown that the relationship between ankle strength and gait speed can vary depending on the walking speed. This study agrees with previous studies indicating a positive relationship between the ankle dorsiflexion strength and preferred gait speed [56,57]. These results suggest that the level of the lower extremity muscle strength used during walking can be adjusted according to the walking speed of the elderly to maintain a comfortable and stable walking speed in daily life. This suggests that improving dorsiflexion strength, which is the applied strength, may be relatively important. Accordingly, it is judged that the elderly require a strategy to detect and treat related factors according to their walking speed in daily life. 

Second, increased MV_ML and RMS_ML, which are related to unstable postural control processing in the static postural control of the elderly, predict functional decline in walking, which is characterized by spatiotemporal gait parameters such as gait speed, stance duration, swing duration, double support duration, and stride length. It has been shown that impaired control in static balance is related to an increased risk of falls in the elderly caused by gait instability [58]. Lopes et al. [59] reported that the COP area showed a negative relationship with stride length in stroke patients with hemiplegia. It has been shown that an unstable static posture may reflect a decrease in walking ability. Hurt et al. [60] compared dynamic balance and step width during gait in older and younger adults, and assessed the relevance of the variables. It was reported that the center of gravity position and acceleration and step width showed a strong correlation. It was also reported that the change in the center of gravity position of the trunk was consciously reduced to control the dynamic stability of gait. The results of this study show that gait performance can be predicted by evaluating postural control in a general static state, as well as the dynamic stability of the torso measured during walking.

Third, the increase in body fat in the elderly is a factor that determines the gait speed and magnitude of the mediolateral movement of the lower body, defined as the swing width during gait. As people age, their muscular mass decreases and their body fat increases [61]. An increase in body fat mass due to reduced activity is a major risk factor indicative of a reduced functional state and disability. It causes muscle atrophy and increases the tension applied to joints and muscles, resulting not only in difficulty in walking, but also decreased ability to perform movements and functions such as balance and getting up from a chair [62,63]. In particular, obese elderly individuals with high body fat showed reduced gait function due to slower speed, wide step widths, and short step lengths when walking compared to elderly individuals of normal weight; however, these characteristics are also considered as strategies to increase the stability of walking [64,65]. This study confirmed that high body fat in the elderly minimally cause a decrease in walking ability through the negative relationship between body fat mass and stride length. However, muscle mass does not affect gait performance because an increase in muscle mass does not necessarily increase muscle strength [66]. Previous studies have shown that reducing body fat rather than increasing muscle mass, and placing importance on muscle functions, such as muscle strength, can be effective in preventing falls. These results agree with those reported previously [67,68,69]. Accordingly, to increase stride length and gait speed stably during walking, the elderly require a strategy to improve muscle function and reduce body fat through exercise and rehabilitation programs. 

Fourth, ankle plantar flexion strength affected walking speed variability. The elderly generally show reduced plantar flexion strength compared to younger adults due to aging [1]. Previous gait studies related to aging have mainly shown that elderly people with low muscle strength increasingly compensate for the decreased gait function when walking compared to young adults; this is conducted to ensure that the gait speed can be maintained, but its variability (standard deviation) is relatively high [69]. Allet et al. [70] suggested that for the elderly population suffering from diabetes, the smaller the difference in the variability of the gait cycle time between the ground (which is divided into stable and unstable ground), the stronger the maximum isometric muscle strength. It was shown that lower extremity strength is an important factor for stable walking. In this study, the effect of plantar flexion strength on gait speed variability in the elderly necessitates the development of a strategy to improve the ability to control gait through plantar flexion strength training. In addition, the ankle plantar flexion strength, used as the lower extremity muscle strength variable from among the independent variables for the spatial variables of gait, was found to account for a large part of the energy required to maintain stability when moving the lower extremities forward during walking. It can be predicted that the strong ankle muscle strength to obtain forward propulsion during walking could result in a forward-stabilized gait pattern. The results of this study show similarity with those of previous studies that demonstrated that the lower extremity muscle strength is an important factor in determining gait flexibility in the elderly [7], and it has been shown that lower extremity strength is closely related to falls. 

### Imitation of Study

This study has several limitations. No assessment of lower limb range of motion and pain was conducted, and it is known that this can influence gait. Each movement was performed three times, and the maximum value was used. However, a break of 30 s could be relatively short and can affect the results. Finally, the level of effort could be an influential factor that can be different for the elderly compared with young adults [71]. However, we did not quantify this factor in the present study.

## 5. Conclusions

This study reveals that the walking ability of the elderly can be influenced by multiple factors, such as improvement in ankle dorsiflexion strength and mediolateral direction posture control ability. Most importantly, ankle dorsiflexion strength was the most influential factor affecting gait speed. However, the strength of plantar flexion of the ankle was the only variable affecting gait variability. In future studies, it will be necessary to confirm whether the improvement in the influencing factors leads to a substantial improvement in gait. The verification of how much the appropriate treatment for factors affecting gait ability and variability actually helps maintain and improve mobility in the elderly determines the direction of the rehabilitation exercise. Training and rehabilitation programs aimed at preventing falls and improving gait performance and variability in the elderly may benefit from these results. The importance of these variables in the evaluation of walking function can help improve the effectiveness of exercise programs.

## Figures and Tables

**Figure 1 medicina-58-01640-f001:**
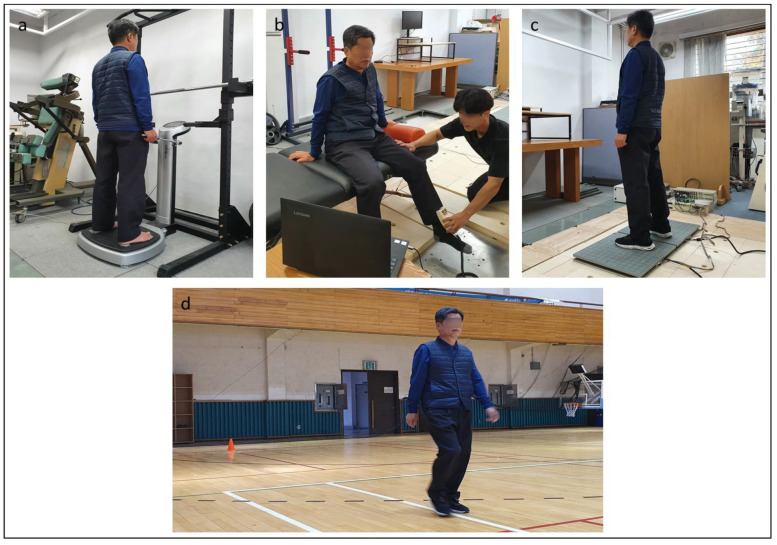
Example of experimental protocol: (**a**) body composition measurement; (**b**) lower extremity muscle strength test; (**c**) static balance test; (**d**) the 6-minute walk test.

**Table 1 medicina-58-01640-t001:** Flowchart showing the experimental design of this study.

Assessed for Eligibility (*n* = 101)
Excluded participants who did not meet the inclusion criteria (*n* = 18)
Excluded participants who declined to participate in the experiment (*n* = 3)
Did not receive allocated assessment in elderly male group (*n* = 2)
Final analyzed (*n* = 78)

**Table 2 medicina-58-01640-t002:** Physical characteristics of the subjects and information on the measured muscle strength, static posture control, body composition, and gait performance.

	The Elderly (*n* = 78)			
Parameters	Totals (*n* = 78)	Males (*n* = 40)	Females (*n* = 38)	*p*-Value
Age (years)	75.69 (5.95)	74.73 (6.92)	76.71 (4.59)	0.14
Height (cm)	160.95 (8.56)	167.47 (5.67)	154.09 (4.92)	<0.01
Weight (kg)	64.66 (9.31)	68.86 (8.45)	60.24 (8.12)	<0.01
Body composition				
Soft lean mass (kg)	41.97 (7.40)	47.66 (5.42)	35.99 (3.37)	<0.01
Body fat mass (kg)	20.22 (5.88)	18.47 (5.05)	22.08 (6.17)	0.01
BMI (kg/m^2^)	24.98 (3.12)	24.57 (2.72)	25.42 (3.47)	0.23
Lower extremity strength				
Flexion iliopsoas (N)	189.33 (61.00)	221.20 (61.20)	155.77 (39.23)	<0.01
Extension gluteus (N)	150.78 (67.14)	188.74 (63.37)	110.81 (44.17)	<0.01
Abduction gluteus (N)	129.21 (42.51)	151.69 (43.43)	105.54 (25.50)	<0.01
Flexion hamstring (N)	99.45 (41.34)	115.32 (35.13)	82.74 (41.18)	<0.01
Extension femoris/vastus (N)	217.67 (60.94)	242.34 (60.19)	191.70 (50.59)	<0.01
Plantar flexion (N)	168.36 (57.78)	197.76 (53.80)	137.42 (44.50)	<0.01
Dorsiflexion (N)	139.33 (43.26)	159.40 (40.91)	118.20 (35.16)	<0.01
Static balance				
RMS_AP (mm/s)	4.70 (1.27)	4.92 (1.34)	4.47 (1.18)	0.12
RMS_ML (mm/s)	2.21 (0.90)	2.13 (0.81)	2.30 (0.98)	0.40
MV_AP (mm/s^2^)	9.30 (2.01)	9.12 (2.12)	9.48 (1.90)	0.42
MV_ML (mm/s^2^)	7.64 (2.18)	7.01 (2.18)	8.30 (1.20)	0.01
Gait performance				
Speed (m/s)	1.22 (0.20)	1.26 (0.19)	1.18 (0.20)	0.03
Stance duration (%)	61.31 (2.71)	61.19 (2.95)	61.43 (2.47)	0.41
Swing duration (%)	38.69 (2.71)	38.81 (2.95)	38.57 (2.47)	0.41
Double support duration (%)	22.19 (4.49)	21.94 (5.16)	22.45 (3.70)	0.24
Stride length (m)	1.22 (0.18)	1.29 (0.18)	1.15 (0.16)	<0.01
Swing width (m)	0.04 (0.02)	0.04 (0.01)	0.04 (0.02)	0.35
Gait variability				
Speed CV (%)	4.74 (1.51)	4.44 (1.39)	5.06 (1.59)	0.06
Stance duration CV (%)	2.48 (1.00)	2.43 (0.98)	2.53 (1.04)	0.64
Swing duration CV (%)	3.94 (1.54)	3.83 (1.50)	4.06 (1.61)	0.50
Double support duration CV (%)	10.51 (4.57)	11.02 (3.89)	9.98 (5.19)	0.31
Stride length CV (%)	3.75 (1.25)	3.49 (1.20)	4.03 (1.26)	0.56
Swing width CV (%)	26.84 (16.87)	28.48 (9.33)	25.12 (22.25)	0.38

Data are presented as mean (SD); the *p*-value indicates the difference between genders. Abbreviations: RMS, root mean square; MV, mean velocity; AP, anterior–posterior; ML, mediolateral; CV, coefficient of variation.

**Table 3 medicina-58-01640-t003:** Summary of results of stepwise regression analysis of gait performance in the elderly (*n* = 78).

Dependent Variables	Independent Variable				(95% CI)
	R^2^	β	Predictors	*p* Value	
Speed(s)	0.291	0.311 −0.291 −0.216	Dorsiflexion Body fat mass RMS_ML	0.003 ** 0.005 ** 0.037 *	(0.112, 0.510) (−0.490, −0.092) (−0.418, 0.013)
Stance duration (%)	0.322	0.322	RMS_ML	0.004 **	(0.106, 0.538)
Swing duration (%)	0.322	−0.322	RMS_ML	0.004 **	(−0.538, −0.106)
Double support duration (%)	0.438	0.226 −0.309 −0.263	RMS_ML Dorsiflexion MV_ML	0.038 * 0.007 ** 0.019 *	(0.013, 0.438) (−0.532, −0.087) (−0.482, −0.044)
Stride length (m)	0.653	0.283 −0.245 0.263 −0.228	Height RMS_ML Plantar flexion Body fat mass	0.009 ** 0.009 ** 0.013 * 0.018 √	(0.072, 0.494) (−0.426, −0.063) (0.058, 0.469) (−0.415, −0.041)
Swing width (m)	0.516	0.308 0.359 0.320 0.296	Dorsiflexion MV_ML Body fat mass Height	0.010 ** 0.002 ** 0.005 ** 0.015 √	(0.077, 0.540) (0.138, 0.581) (0.101, 0.539) (0.058, 0.533)

Abbreviations: MV, mean velocity; ML, mediolateral; RMS, root mean square; CI, confidence interval. * *p* < 0.05, ** *p* < 0.01, √ *p* value is not significant by Holm correction.

**Table 4 medicina-58-01640-t004:** Summary of results of stepwise regression analysis of gait variability in the elderly (*n* = 78).

Dependent Variables	Independent Variable				(95% CI)
	R^2^	β	Predictors	*p* Value	
Speed CV (%)	0.426	−0.396 0.233	Plantar flexion RMS_AP	<0.001 *** 0.031 √	(−0.607, −0.185) (0.022, 0.443)
Stance duration CV (%)				n.s	
Swing duration CV (%)				n.s	
Double support duration CV (%)				n.s	
Stride length CV (%)	0.436	−0.436	Plantar flexion	<0.001 ***	(−0.642, −0.231)
Swing width CV (%)				n.s	

Abbreviations: RMS, root mean square; CV, coefficient of variation; CI, confidence interval; n.s., not significant; *** *p* < 0.001, √ *p* value is not significant by Holm correction; AP, anteroposterior.

## Data Availability

The datasets used during the current study are available from the corresponding author upon reasonable request.

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
