# Peer review of "Determination of the Predictors with the Greatest Influence on Walking in the Elderly"

_medicina, 2022, doi:10.3390/medicina58111640_

Round 1
Reviewer 1 Report
In "Material and methods" it is described that all the participants are over 60 years of age (line 115), but it is verified that all of them are over the age of 70 years.
The Height (cm) and the Weight (kg) showed significant differences between the groups (both <0.01), as did the Soft lean mass (<0.01) and the Body fat mass (0.01). All these variables could influence the results. However, BMI did not show a significant difference. Don't you think these variables could have had more importance in the differences found?
This study is consistent with previous studies in indicating a positive relationship between ankle dorsiflexion strength and preferred walking speed. Wasn't the type of footwear that each of the participants took out analyzed in the "Methods" section?
Could any respiratory disease, such as COPD or dyspnea on exertion, not considered as an exclusion criterion in the study, influence a lower walking performance? During the 6-minute walk test, were dyspnea and fatigue measured?
Some of the bibliographic citations are double spaced and not aligned.
Author Response
Thank you for your all commments. Now we uploaded the revised version here.
Please see the attachment

Reviewer 2 Report
This study aimed to determine the variable that is the most influential predictor on walking and gait performance in the elderly. These are my comments and suggestions:
Abstract:
Line 12: "n." Please correct.
Abstract is nicely written with clearly stated the purpose of the study.
Introduction:
The Introduction section is well written and you mention the factors that affect walking in the elderly.
Line 98: "Previous studies ..." Change paragraph.
Line 107 - 109: "We hypothesised ..... - ....... gait variability." There is no need for that part. A suggestion is to delete it.
Line 109-110: This can be moved to the conclusion.
Materials and Methods:
Here is a misunderstanding in the text between the Table 1 and Figure 1. Please correct it.
Line 115 and Line 274: You mention over the age of 60. However, the average age of the participants was 75 years.
2.2.1. Lower extremity muscle strength test
Line 154-155: "The lower .......- ....... [27-29]". What do you mean?
You should provide more information about that part. More specifically,
- You mention 7 measurement. How did you measure these muscle groups? You should provide more information about the methodological approach (start of the movement etc).
- In the Table 1 the results of the lower extremity strength are very high (i.e. 242.34 (60.19) N for males and 191.70 (50.59) N for females. How is that possible, considering the age of the participants?
Also, you should provide more relevant research about the reliability of the microFET3. Many researches have used an isokinetic dynamometer. Why not you?
Results:
Results chapter is adequate.
Discussion:
Line 284: How did you measure the dorsiflexion?
Line 347: How did you measure the ankle plantar flexion?
Author Response

(The authors gave the same response as above.)

Round 2
Reviewer 2 Report
I am satisfied with the improvements and corrections of the manuscript.
Author Response
Thank you for your review.
